# Group-Based Trajectory Model to Assess Adjuvant Endocrine Therapy Adherence Pattern in HR-Positive Breast Cancer: Results from Rio Grande Valley Patients [note 1]

**DOI:** 10.3390/healthcare13151777

**Published:** 2025-07-22

**Authors:** Bilqees Fatima, Phillip Shayne Pruneda, Parasto Mousavi, Rheena Sheriff, Ronnie Ozuna, Meghana V. Trivedi, Susan Abughosh

**Affiliations:** 1Department of Pharmaceutical Health Outcomes and Policy, College of Pharmacy, University of Houston, Houston, TX 77204, USA; bfatima5@uh.edu (B.F.); pmousavi@uh.edu (P.M.); 2Department of Pharmacy Practice and Translational Research, College of Pharmacy, University of Houston, Houston, TX 77204, USA; shaynepruneda@gmail.com (P.S.P.); mtrivedi@uh.edu (M.V.T.); 3Doctors Hospital at Renaissance, Edinburg, TX 78539, USA; rheenasheriff@gmail.com; 4Rio Grande Valley, Doctors Hospital at Renaissance, College of Pharmacy, University of Houston, Houston, TX 77204, USA; rozuna@uh.edu

**Keywords:** oral endocrine therapy, group-based trajectory model, breast cancer, adherence

## Abstract

**Background/Objectives**: Adherence to oral endocrine therapy (OET) is essential to reduce recurrence but is predominantly lower among underserved patients, leading to worse health outcomes. We aimed to depict longitudinal patterns of OET adherence using group-based trajectory modeling (GBTM) and identify predictors associated with each adherence trajectory. **Methods**: A single-center, retrospective study was conducted to analyze data from women 18 years or older with metastatic breast cancer who initiated with an OET and were treated from January to December 2022. Adherence was measured using a proportion of days covered (PDC > 80%) for 12 months. Binary monthly indicator of PDC was incorporated into GBTM. Four models were generated by changing the number of groups from 2 to 5, using a 2nd-order polynomial function of time. A multinomial logistic regression model was run to evaluate the predictors of non-adherence trajectories, and “adherence” was considered the reference group. **Results**: A total of 346 women had a (mean age of 60) years; 93% were Hispanic or of Mexican origin; 90% were taking aromatase inhibitors (AIs), with an endocrine therapy of 1.05 years. Three trajectories of adherence to GBTM were identified: a gradual decline in adherence (*n* = 88, 25.5%), improving suboptimal adherence (*n* = 106, 30.6%), and adherent (*n* = 152, 43.9%). Multinomial logistic regression analysis showed that significant predictors are diabetes (odds ratio (OR), 2.96; 95% confidence interval (CI), 1.57–5.57) and fewer years of therapy (OR, 2.96; 95% CI, 1.57–5.57). Suboptimal adherence among RGV patients receiving OET, with approximately 56% following a non-adherent trajectory. **Conclusions**: Suboptimal adherence among RGV patients receiving OET, with approximately 56% following a non-adherent trajectory. Significant predictors should be considered when designing targeted interventions.

## 1. Introduction

Breast cancer is the most common cause of cancer death among women worldwide [1,2]. In the United States in 2023, there were 297,790 new cases of female breast cancer and 43,170 estimated related deaths [3]. Approximately 70% of these instances are hormone (estrogen and/or progesterone) receptor-positive (HR+). HR+ breast cancer has a high survival rate, in part due to the efficacy of oral endocrine therapy (OET), which comprises selective estrogen receptor modulators, such as tamoxifen and aromatase inhibitors. OET is a key component of the treatment of both early and advanced-stage HR+ breast cancer [4,5]. Non-adherence to oral endocrine therapy has been associated with the risk of recurrence and mortality, which ultimately leads to increased healthcare costs [6]. Five years of tamoxifen therapy can substantially reduce the 15-year risk of breast cancer recurrence by up to 40% and decrease mortality by 30% in premenopausal women [7]. For postmenopausal women, aromatase inhibitors are particularly effective, as they significantly lower estrogen levels and can reduce the risk of recurrence by approximately 30% [8]. Together, these therapies represent critical advancements in long-term management and survival outcomes for breast cancer patients [9].

Despite the clinical benefits of OET, numerous studies have continuously reported less-than-ideal adherence rates. For instance, a cohort study in 2010 found that only 49% of U.S. patients were adherent to OET during the 5-year treatment period [10]. A systematic review of 29 studies in 2012 reported adherence rates ranging from 41% to 72%, with discontinuation rates between 31% and 73% [11]. More recently, a 2021 study using the SEER database indicated an average 1-year adherence rate of 87%, but this dropped to 65.2% over 5 years [12]. The adherence rates are even lower for the extended 10-year OET regimen [13]. These findings underscore the challenges in achieving optimal adherence in real-world settings. Among underserved populations, non-adherence is particularly prevalent, especially among patients with lower socioeconomic status [14].

The Rio Grande Valley (RGV) in Texas is a region characterized by a predominantly Hispanic population with high levels of poverty and limited access to healthcare services [15]. This region has a population of around 1.3 million, with over 85% identifying as Hispanic or Latinx. Approximately 30% of the residents live below the poverty line, and approximately 40% lack health insurance. This population experiences significant health disparities [16], and these disparities are exacerbated by socioeconomic challenges such as financial constraints, lack of health insurance, and barriers to accessing care, including transportation issues and a shortage of healthcare providers [17].

The use of group-based trajectory modeling (GBTM) on longitudinal data offers a method for analyzing the natural progression and treatment effects on adherence behavior, by depicting the various patterns of adherence behaviors that naturally occur over time [18]. While conventional approaches like proportion of days covered (PDC) mainly evaluate the amount of medication used in a given period, GBTM identifies unique adherence profiles as well as predictors by categorizing patients into clusters with similar adherence trajectories [19,20]. GBTM allows for the detection of distinct behavioral patterns over time, which is especially useful for tailoring adherence interventions in a diverse population. This study was conducted to evaluate OET adherence using GBTM in a population of breast cancer patients at a multispecialty hospital in the Rio Grande Valley, primarily of Hispanic origin. Identifying prominent adherence trajectories and patient characteristics associated with these trajectories in this underserved population can provide valuable insight into the medication-taking behavior among this underserved patient population and guide the development of tailored interventions to improve endocrine therapy (OET) adherence among Hispanic breast cancer patients in the RGV.

## 2. Materials and Methods

### 2.1. Study Design and Data

This single-center, retrospective study took place at a single hospital site in the Rio Grande Valley with a primarily Hispanic population and a median household income lower than the state average. The study extracted patients with at least one OET dispensed record from January 2022 through December 2022 from the Cerner Electronic Health Record (EHR) system. The dispense history was extracted to determine whether the patients had an OET supply on hand starting from October 2021.

### 2.2. Study Cohort

The study encompassed all women 18 years or older with a breast cancer diagnosis who are currently on appropriate doses of OET. Women taking OET for prevention because of ductal carcinoma in situ (DCIS), lobular carcinoma in situ (LCIS), or ductal or lobular hyperplasia were classified together as stage 0/prevention. Women who were not on appropriate doses of OET (tamoxifen 20 mg once daily, anastrozole 1 mg once daily, letrozole 2.5 mg once daily, and exemestane 25 mg once daily), discontinued OET due to severe side effects, or were on OET for reasons other than BC prevention or treatment were excluded from this study.

### 2.3. Measures

#### 2.3.1. Main Outcome

Adherence trajectories of adjuvant oral endocrine therapy were the primary outcome of interest. The results were measured by monthly PDC over 1-year period, converted into a binary indicator of adherence for each month, and then incorporated into GBTM.

#### 2.3.2. Variable of Interest

Sociodemographic Variables

We investigated the adherence to OET over time and the factors associated with adherence trajectories and collected data on patient demographics such as birth date, gender, and ethnicity.

Clinical Variables

The clinical variables collected included the following: vitals (body weight, height, body mass index (BMI), medical history including comorbid conditions (diabetes, hypertension, hyperlipidemia, depression), clinical cancer stage and pathological cancer stage at diagnosis or the start of any treatment (chemotherapy or OET), ER/PR/HER status, current endocrine therapy for the study period (name, dose, frequency), date of first prescription for current treatment, date of current treatment initiation, other cancer therapy use (CDK4/6 inhibitor, fulvestrant, LHRH/GnRH agonists), refill dates and quantity throughout the study period, date and reason of discontinuation, the number of months’ supply filled in-person and via home delivery, and date and regimen of original OET initiation.

### 2.4. Data Management

The electronic health records were used to obtain patients’ prescription refill data, which includes patient demographics and clinical variables, dosing, frequency, dispense date, quantity, and reason for discontinuation. Initially, pilot data of thirty patients was collected to review any necessary edits in the data collection form and validate the consistency of data collection by two investigators. Licensed pharmacists addressed any queries associated with data collection requirements. The final audit consisted of a random selection of patients from the data collection sheet and performed a thorough review to ensure data integrity.

### 2.5. Adherence Measurements

#### 2.5.1. Binary Adherence

The adherence was quantified using the proportion of days covered (PDC) from January to December 2022. The prescription refill data was obtained from the outpatient pharmacies’ Cerner Electronic Health Record (EHR) system; however, it does not provide information on whether patients picked up the medications. Medication fills history may not fully reflect adherence. However, repeated prescription refills, especially when patients are paying a copay, indicate that patients are using their medications and correlate with adherence, which is why it is an accepted measure of adherence [21]. Adherence was estimated by PDC for each consecutive 30-day period as the number of days “covered” by a prescription and divided by the total number of days in the measurement period, adjusting for medication overlap. Then, we created the binary indicator of adherence (PDC ≥ 80%) to measure monthly adherence for 12 months.

#### 2.5.2. Group-Based Trajectory Model

The 12 binary indicator of monthly PDC was then incorporated into GBTM. Four models were developed by changing the number of groups from 2 to 5, using a 2nd and 3rd-order polynomial function of time. Multiple logistic regression equations were used simultaneously by the group-based trajectory model to estimate the membership probability among each trajectory group and the probability of being adherent to OET as a function of time. The model parameters were estimated using maximum likelihood estimates. The final model was selected based on the Bayesian information criterion (BIC) (Appendix A Table A1), clinical relevance, and a minimum sample size of 5% required in each trajectory group. (Appendix A Table A2) [19,22]. Once identified, the trajectories were characterized based on a multinomial logistic regression model to assess the predictors of non-adherence trajectories, and “adherence” was considered the reference group.

#### 2.5.3. GBTM Validation

The model performance evaluation of the selected 3-group trajectory model was assessed by following the Nagin model adequacy criteria [23]. To determine how well the model fits with the data, the following goodness of fit measures have been suggested: (1) average posterior probability of assignment (AvePP) values more than or equal to 0.7 for all trajectory groups reflect high posterior probabilities of assigning observations to their groups. Similarly, (2) odds of correct classification (OCC), which is estimated as the proportion of the odds of correct classification based on the maximum probability classification rule and the estimated proportion of class members, the recommended OCC value is 5 or more for all groups of the trajectories, which indicates that the model fits the data well [22]. The model fit indices of the selected three-group model achieved an AvePP ≥  0.85 and OCC values >  5.0 for each trajectory, indicating good classification accuracy (Appendix A Table A2).

### 2.6. Statistical Analysis

The chi-square test for categorical variables and the ANOVA test for continuous variables were used to identify the group differences in patient characteristics between adherence trajectories. Next, using a multinomial logistic regression model, we evaluated the outcome of interest (adherence trajectories with “adherence” trajectory as the reference group) and its association with variables of a priori strong clinical interest. Covariates included in the model were age (<60 vs. ≥60 years), ethnicity (Hispanic/Latino vs. non-Hispanic/Latino), BMI (≤24.9, 25–29.9, ≥30), oral endocrine therapy (tamoxifen, AIs), stage of cancer (stage I–III), ER/PR/HER2, years on therapy (≤1 year, ≥2 years), and comorbidities such as depression, diabetes, hyperlipidemia, and hypertension. Statistical analysis was performed using Statistical Analysis System (SAS) version 9.4 (SAS Institute, Cary, NC, USA) with a statistical significance defined as 2-sided *p* < 0.05. “Proc Traj,” an add-on user-written program, was used for trajectory modeling.

## 3. Results

### 3.1. Baseline Characteristics

A total of 346 patients were included in the final analysis based on the inclusion and exclusion criteria. Patient demographic and clinical characteristics are detailed in Table 1. Most women were above the age of 60 (*n* = 175, 51%), 93% were Hispanic or of Mexican origin (*n* = 322), and most of the patients were classified as obese (*n* = 184, 53.18%).

The most significant comorbidities were diabetes (*n* = 116, 34%), followed by hyperlipidemia (*n* = 168, 48%). Other comorbidities found in this patient population were hypertension (*n* = 200, 58%) and depression (*n* = 53, 15%). Most patients diagnosed were with stage I (*n* = 163, 47%) and stage II (*n* = 127, 37%), followed by stage III (*n* = 56, 16%). AIs were the more common OET (*n* = 311, 90%) compared to tamoxifen (*n* = 35, 10%). The mean (SD) follow-up year of oral endocrine therapy was 1.05 (1.12) for this patient cohort.

### 3.2. Adherence Trajectories of OET

Three groups were selected as the final model validated by using Nagin’s criteria for model validation in addition to Bayesian information criteria, clinical relevance, and a 5% minimum membership requirement. Three distinct trajectories of OET adherence were selected as a gradual decline in adherence (*n* = 88, 25.5%), improving suboptimal adherence (*n* = 106, 30.6%), and adherent (*n* = 152, 43.9%) (Figure 1). The demographic and clinical characteristics of patients in each trajectory are presented in Table 1.

### 3.3. Predictors Associated with Memberships in Adherence Trajectories

The multinomial logistic regression model showed that having diabetes was significantly associated with the gradual decline trajectory (OR, 2.96; 95% confidence interval (CI), 1.57–5.57). Moreover, a significant predictor associated with improving suboptimal adherence included fewer years of therapy (OR, 0.32; 95% CI, 0.218–0.459). Table 2 summarizes factors associated with suboptimal adherence trajectories.

## 4. Discussion

Ethnicity and socioeconomic status are frequently associated with non-adherence to OET, and several studies demonstrated that non-adherence to OET was higher among patients with lower socioeconomic status [2,24,25,26]. In this study, overall, three groups were identified with distinct trajectories of adherence over 12 months of the follow-up period: “a gradual decline in adherence,” “improving suboptimum adherence,” and “adherent,” which is the reference group in the multinomial logistic regression model. In this study, by using GBTM, we identified a larger proportion of patients following an adherent trajectory (*n* = 152, 43.9%) compared to a 12-month PDC binary adherence categorization, which identified only (*n* = 122, 35.3%) as adherent using a threshold of PDC ≥ 0.8. Further analysis revealed that 30 patients classified as adherent by GBTM but not by the binary model had PDC ranging from 0.724 to 0.799, indicating a borderline adherence status closer to the PDC ≥ 0.8 cutoff. In the clinical setting, these near-threshold patients could be managed well to prevent further decline in their adherence. This difference between the GBTM and the binary adherence method highlights the methodological differences, as the binary indicator results in a major loss of information and does not account for the dynamic nature of adherence, especially for long-term medications, while GBTM can identify the homogenous groups of patients based on the probabilities of adherence as a function of time [19,27].

We observed that a proportion of two-thirds (56.1%) of women followed sub-optimal adherence trajectories, with only 43.9% of the patients consistently following adherent trajectories (PDC > 80%) in the final selected model. A French cancer cohort study reported trajectories of adherence to adjuvant endocrine therapy for 5 years. The proportion of patients that belonged to the most adherent group was approximately similar to ours (43% vs. 44%). However, this study has identified six trajectories of adherence [28]. In addition, Winn et.al. reported five trajectories of adherence by using Surveillance, Epidemiology, and End Results-Medicare data, and approximately 56% of patients in the high adherence trajectory, which is higher than our study [29]. This might be due to a longer follow-up period in the study. However, none of the studies reported trajectories of OET adherence among underserved patient populations.

Moreover, the literature revealed that the findings of our study are consistent with numerous previously published studies on trajectories of adherence to OET [28,29,30]. Studies using GBTM for assessing OET adherence have found 5–6 trajectories [27,28,31]. However, our study identified three trajectories, which could be due to a smaller sample size and shorter follow-up time. The assessment of model adequacy is a significant step in GBTM [23]. To determine the optimal number of adherence trajectories, we compared models 2–5 based on BIC of where smaller absolute values represent better model fit, clinical relevance, and a minimum sample size of 5% required in each trajectory group. In addition, we have used Nagin model adequacy criteria and assessed the goodness of fit measures, including AvePP ≥ 0.85, and OCC values  >  5.0 for each trajectory, which indicates good classification accuracy. This study descriptively reported the improving sub-optimal adherence to OET as a major trajectory (35%) in an underserved Hispanic patient population, while this trajectory reported a small number of patients in other GBTM of OET adherence studies [27,31]. This increase in the number of patients in our study in this trajectory group could be due to financial toxicity and adverse events associated with OET use in this specific patient.

In comparison to other studies, our study projected that diabetes among breast cancer patients was a significant predictor of lower adherence. Diabetes is considered one of the most prevalent comorbidities in breast cancer patients and affects up to 30% of them [32]. In previous studies, the effect of comorbidities on patient adherence varied according to the number of comorbid conditions, such as patients with more than three comorbidities had the highest rates of non-adherence [33]. In one of the population-based cohort studies, pre-existing diabetes was associated with subpar adjuvant therapies for breast cancer among low-income women who were continuously enrolled in Medicaid claims data, which is linked to the Missouri Cancer Registry [34]. They reported that patients with diabetes had a higher risk of non-adherence to endocrine therapy in the first year of treatment [34], which is significantly relevant to introducing an intervention addressing the issue of suboptimal adherence, especially during the initial year of treatment [35]. Consistent with these studies, we found that factors associated with groups of non-adherent women were comorbid conditions such as diabetes mellitus. Therefore, improving diabetes management during breast cancer treatment is important for low-income women who may have been disproportionately affected by diabetes, as it affects RGV patients (44%) significantly higher than nationwide (27%) [36].

Furthermore, years of therapy have been shown to affect suboptimal adherence to OET. Patients prescribed OET for over two years showed 61% adherence, compared to 39% of patients given therapy for less than 1 year. This finding differs from previous reports in the literature, where non-adherence to OET was associated with more years of therapy [37,38]. A possible reason for better adherence with longer follow-up is related to addressing adherence barriers through continued monitoring and regular follow-up with health care providers, which is essential to improve adherence over time. As the underserved population is unable to adhere to their medications due to numerous systemic barriers such as limited access, financial constraints, limited health literacy, and cultural barriers [39,40]. Future studies could design targeted interventions to improve OET adherence and assess their feasibility and effectiveness. An educational intervention that is culturally sensitive to the cultural norms and addresses barriers specific to this population can be more influential with patients, leading to behavioral change, better adherence, and ultimately enhancing health outcomes.

### 4.1. Strengths

Our study has several strengths and could be a meaningful addition to the current literature on OET adherence among the underserved patient population. Our study has identified a major area of unmet clinical need to improve OET adherence for BC prevention among a minority population, a largely ignored area of research until now. Our study highlights the critical unmet need and can be used to design targeted future interventions that would be appropriate to the predominant Hispanic patient population.

### 4.2. Limitations

There are certain limitations to this study, such as the Cerner electronic health record system not being able to capture medication refills processed at the pharmacy; however, it does not provide information on whether patients picked up the medications. Consequently, it might overestimate the adherence rates. Refill data could be misclassified if adherent patients are considered non-adherent or vice versa. Moreover, residual uncontrolled confounding may exist as we did not account for a few sociodemographic factors, such as education level, insurance status, or means of transportation, which could confound sub-optimal adherence rates. Additionally, the study measured adherence for only 12 months at any time during patients’ OET therapy. A recommended approach would be to assess adherence from initiating OET for at least five years or a longer treatment duration. In several subgroups, a smaller sample size could have biased our analysis. Studies that represent the national cohort are needed to address the impact of ethnicity and socioeconomic status on OET adherence. Despite these limitations, findings of our GBTM reported that only 43% of the sample of predominantly Hispanic patients with low socioeconomic status followed an optimal adherent trajectory, highlighting the need for targeted interventions to mitigate the specific barriers and enhance OET adherence in this patient population, which is crucial to prevent recurrence.

## 5. Conclusions

In summary, two-thirds of women diagnosed with metastatic BC who initiated OET had a suboptimal trajectory of adherence over the 12 months of treatment. GBTM is important for identifying longitudinal OET adherence patterns among HR+ BC patients. A better understanding of these sub-optimal adherence trajectories for identifying and managing them for patients, physicians, and other healthcare team members could enhance OET adherence and prevent recurrence and mortality among metastatic BC patients. Integration of diabetes management led by nurse practitioners into the multidisciplinary approach for women with breast cancer should be added for comprehensive management to mitigate special needs for patients with both diabetes and breast cancer [41]. In addition, future studies should investigate interventions to improve OET adherence among minority patients with lower socioeconomic status, patients with diabetes, and in the early years of therapy that might hinder adherence. Also, future studies among ethnically underserved patient populations should investigate barriers associated with OET non-adherence.

## Figures and Tables

**Figure 1 healthcare-13-01777-f001:**
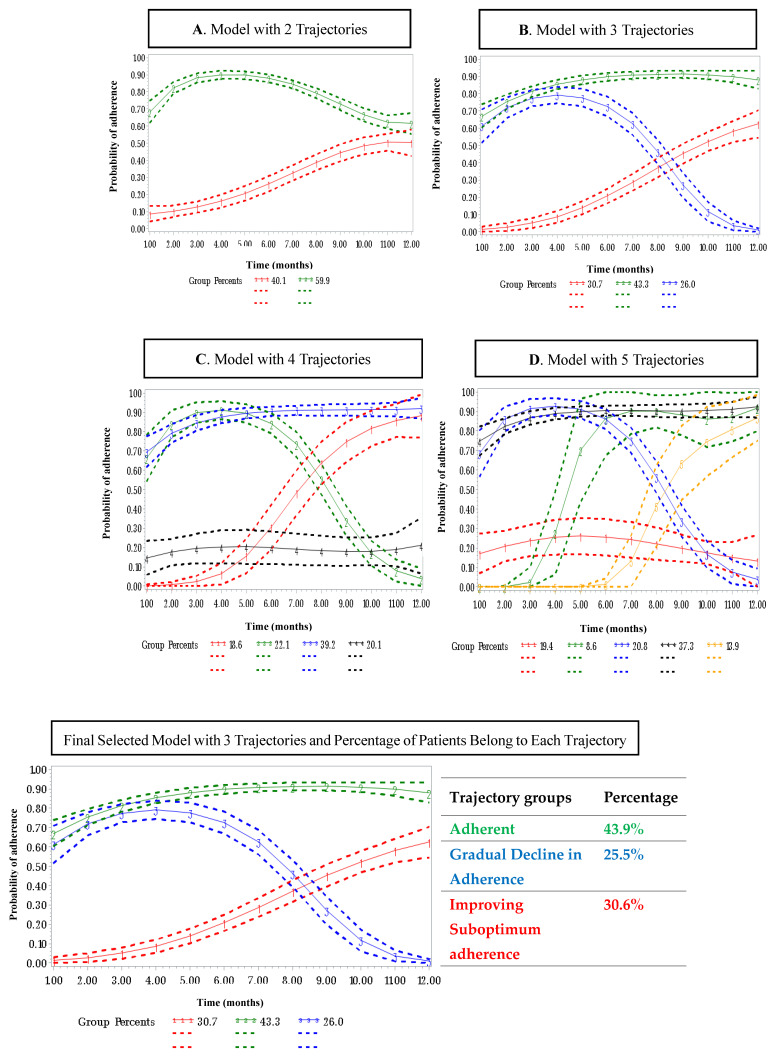
Trajectories of oral endocrine therapy (OET) over 12 months (results of the 5 group-based trajectory models second-order polynomial function of time, which included 2 to 5 trajectories): (**A**) 2 trajectories; (**B**) 3 trajectories; (**C**) 4 trajectories; and (**D**) 5 trajectories. Note: solid lines represent the predicted probability of adherence in each group; the observed proportion is plotted with dotted lines.

**Table 1 healthcare-13-01777-t001:** Baseline characteristics of the study population.

Variable	All Patients (*n* = 346)	Improving Suboptimum Adherence (*n* = 106)	Adherent (*n* = 152)	Gradual Decline in Adherence *(n* = 88)	*p* Value
**Age (years), *n* (%)**					0.029
<60	171 (49.42)	63 (59.43)	72 (47.37)	36 (40.91)	
≥60	175 (50.58)	43 (40.57)	80 (52.63)	52 (59.09)	
**Ethnicity, *n* (%)**					0.136
Hispanic	322 (93.06)	100 (94.34)	137 (90.13)	85 (96.59)	
Not Hispanic	24 (6.94)	6 (5.66)	15 (9.87)	3 (3.41)	
**BMI**					0.533
Obese	184 (53.18)	61 (57.55)	79 (51.97)	44 (50.00)	
Non-obese	162 (46.82)	45 (42.45)	73 (48.03)	44 (50.00)	
**Endocrine therapy, *n* (%)**					0.868
Aromatase inhibitors	311 (89.88)	94 (88.68)	137 (90.13)	80 (90.91)	
Tamoxifen	35 (10.12)	12 (11.32)	15 (9.87)	8 (9.09)	
**Clinical Stage of Cancer, *n* (%)**					0.67
Stage 1	163 (47.11)	50 (47.17)	68 (44.74)	45 (51.14)	
Stage 2	127 (36.71)	37 (34.91)	62 (40.79)	28 (31.82)	
Stage 3	56 (16.18)	19 (17.92)	22 (14.47)	15 (17.05)	
**ER_PR_HER2, *n* (%)**					0.60
HER2+	56 (16.18)	20 (18.87)	24 (15.79)	12 (13.64)	
HER2-	290 (83.82)	86 (81.13)	128 (84.21)	76 (86.36)	
**Comorbidities**					
**Diabetes, *n* (%)**					0.0001 *
Yes	116 (33.53)	30 (28.30)	39 (25.66)	47 (53.41)	
No	230 (66.47)	76 (74.70)	113 (74.34)	41 (46.59)	
**Hypertension, *n* (%)**					0.07
Yes	200 (57.80)	57 (53.77)	83 (54.61)	60 (68.18)	
No	146 (42.20)	49 (46.23)	69 (45.39)	28 (31.82)	
**Hyperlipidemia, *n* (%)**					0.04 *
Yes	168 (48.55)	44 (41.51)	72 (47.37)	52 (59.09)	
No	178 (51.45)	62 (58.49)	80 (52.63)	36 (40.91)	
**Depression, *n* (%)**					0.92
Yes	53 (15.32)	17 (16.04)	22 (14.47)	14 (15.91)	
No	293 (84.68)	89 (83.96)	130 (85.53)	74 (84.09)	
** Number of years of therapy, ** ** *n* ** **(%)**					
≤1 year	136 (39.31)	86 (81.13)	31 (20.39)	19 (21.59)	0.0001 *
≥2 years	210 (60.69)	20 (18.87)	121 (79.61)	69 (78.41)	

* Statistically significant difference *p* value < 0.05 or <0.01.

**Table 2 healthcare-13-01777-t002:** Results of the multinomial logistic regression model.

Variable	Reference	Improving Suboptimum Adherence vs. Adherent			Gradual Decline vs. Adherent		
		OR	95% CI	*p* Value	OR	95% CI	*p* Value
Age							
≥60	<60	0.67	0.35–1.32	0.25	1.07	0.58–1.96	0.81
Ethnicity							
Not Hispanic	Hispanic	0.96	0.27–3.40	0.95	0.41	0.11–1.52	0.18
BMI							
Obese	Non-Obese	1.32	0.71–2.48	0.37	1.00	0.56–1.78	0.98
Endocrine therapy							
Tamoxifen	Aromatase Inhibitors	1.41	0.41–4.07	0.52	1.10	0.40–2.90	0.86
Clinical Stage of Cancer							
Stage 2	Stage 1	0.96	0.521–1.789	0.91	0.64	0.343–1.191	0.15
Stage 3	Stage 1	1.24	0.552–2.820	0.59	1.07	0.472–2.438	0.86
ER_PR_HER2							
HER2+	HER2-	1.61	0.355–7.304	0.84	0.773	0.256–2.337	0.61
Diabetes							
Yes	No	1.40	0.66–2.97	0.38	2.96	1.57–5.57	0.0006 *
Hypertension							
Yes	No	1.82	0.90–3.66	0.09	1.32	0.69–2.53	0.39
Hyperlipidemia							
Yes	No	1.11	0.55–2.22	0.76	1.06	0.56–1.99	0.85
Depression							
Yes	No	1.21	0.50–2.88	0.67	1.19	0.55–2.58	0.65
Number of years of therapy							
≥2 years	≤1 year	0.32	0.218–0.459	0.0001 *	0.80	0.402–1.605	0.535

* Statistically significant difference *p* value < 0.05, or <0.01; abbreviation: odds ratio (OR), confidence interval (CI), estrogen receptor (ER), progesterone receptor (PR), and human epidermal growth factor receptor 2 (HER2).

## Data Availability

The data presented in this study are available on request from the corresponding author.

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
