# Peer review of "Group-Based Trajectory Model to Assess Adjuvant Endocrine Therapy Adherence Pattern in HR-Positive Breast Cancer: Results from Rio Grande Valley Patients [Author-notes fn1-healthcare-13-01777]"

_healthcare, 2025, doi:10.3390/healthcare13151777_

Round 1
Reviewer 1 Report
Comments and Suggestions for Authors
This manuscript addresses an important and understudied area: patterns of adherence to oral endocrine therapy (OET) among underserved, predominantly Hispanic women with HR-positive breast cancer in the Rio Grande Valley (RGV), Texas.
Personally, the use of Group-Based Trajectory Modeling (GBTM) to identify adherence trajectories adds methodological strength and provides nuanced insights beyond traditional binary metrics.
The focus on an underserved Hispanic population is both timely and essential, given well-documented healthcare disparities.
The authors provide compelling evidence that nearly 60% of patients follow suboptimal adherence trajectories, which has clear implications for policy and practice.
While the study fills an important gap, the novelty is somewhat limited by similar prior work using GBTM in other settings. The originality rests more in the population studied than in the methods used.
Methods
Measurement of Adherence: The use of PDC and GBTM is well-justified. However, the limitation of relying on refill records without confirmation of medication intake should be emphasized more.
Trajectory Modeling: The selection of the 3-group model was appropriately justified using BIC, APPA, and OCC. However, more transparency about the actual model fit indices and residual checks would strengthen the methodological section.
- Results
The identification of three adherence trajectories (adherent, gradual decline, improving suboptimal adherence) is meaningful. The finding that diabetes is a strong predictor of the "gradual decline" group is important and actionable.
- Discussion
Contextualization: The discussion is well-informed and relates findings to existing literature. The differences with prior studies are explained plausibly (e.g., sample size, population differences).
- Writing and Structure
The manuscript is well-written with a professional tone.
Minor grammatical and typographical errors should be corrected during copyediting (e.g., occasional awkward phrasing, such as “significantly important”).
- Ethical Considerations
Ethical approval and data usage protocols were clearly reported.
The lack of patient-level consent is acceptable given the retrospective, de-identified nature of the dataset.
Suggestions:
While the study is robust and meaningful, several revisions are recommended to improve clarity, methodological transparency, and discussion depth.
Clarify the timeline for identifying adherence—does “12-month adherence” refer to the first 12 months of therapy, or any 12-month window within ongoing therapy?
Address data limitations: Consider elaborating more critically on how refill data may misclassify adherence.
Expand on GBTM model diagnostics: Include more detailed fit statistics and possibly a sensitivity analysis comparing different group numbers.
Discuss missing data handling: Were there missing data in clinical covariates, and how were they handled?
Enhance discussion of interventions: What kinds of interventions (e.g., educational, pharmacologic, digital health tools) are likely to be feasible and culturally appropriate for this population?
Improve statistical reporting: Tables could benefit from confidence intervals for key baseline statistics to help contextualize variability.
This study offers valuable insight into adherence patterns among Hispanic breast cancer patients using robust analytic methods.
Author Response
Reviewer 1:
We thank the reviewer for thoroughly reviewing the manuscript and providing suggestions for improvement. Point-wise response to each comment is given below. Corresponding changes in the manuscript are highlighted by using track changes.
This manuscript addresses an important and understudied area: patterns of adherence to oral endocrine therapy (OET) among underserved, predominantly Hispanic women with HR-positive breast cancer in the Rio Grande Valley (RGV), Texas.
Personally, the use of Group-Based Trajectory Modeling (GBTM) to identify adherence trajectories adds methodological strength and provides nuanced insights beyond traditional binary metrics.
The focus on an underserved Hispanic population is both timely and essential, given well-documented healthcare disparities.
The authors provide compelling evidence that nearly 60% of patients follow suboptimal adherence trajectories, which has clear implications for policy and practice.
While the study fills an important gap, the novelty is somewhat limited by similar prior work using GBTM in other settings. The originality rests more in the population studied than in the methods used.
Methods
Measurement of Adherence: The use of PDC and GBTM is well-justified. However, the limitation of relying on refill records without confirmation of medication intake should be emphasized more.
Author response: We elaborated in limitation section as one of the limitations, also we have added in the method section with reference that “The prescription refill data was obtained from the outpatient pharmacies Electronic Medical Record system; however, it does not provide information on whether patients picked up the medications. Medication fills history may not fully reflect adherence. However, repeated prescription refills, especially when patients are paying a copay, indicates that patients are using their medications and correlates with adherence, which is why it is an accepted measure of adherence. “(Line-139-144)
Trajectory Modeling: The selection of the 3-group model was appropriately justified using BIC, APPA, and OCC. However, more transparency about the actual model fit indices and residual checks would strengthen the methodological section.
Author response: Added as suggested (Line:171-173)
- Results
The identification of three adherence trajectories (adherent, gradual decline, improving suboptimal adherence) is meaningful. The finding that diabetes is a strong predictor of the "gradual decline" group is important and actionable.
Author response: thanks
- Discussion
Contextualization: The discussion is well-informed and relates findings to existing literature. The differences with prior studies are explained plausibly (e.g., sample size, population differences).
Author response: thanks
- Writing and Structure
The manuscript is well-written with a professional tone.
Minor grammatical and typographical errors should be corrected during copyediting (e.g., occasional awkward phrasing, such as “significantly important”).
Author response: we rewarded phrases throughout the manuscript
Ethical Considerations
Ethical approval and data usage protocols were clearly reported.
Author response: thanks
The lack of patient-level consent is acceptable given the retrospective, de-identified nature of the dataset.
Suggestions:
While the study is robust and meaningful, several revisions are recommended to improve clarity, methodological transparency, and discussion depth.
Clarify the timeline for identifying adherence—does “12-month adherence” refer to the first 12 months of therapy, or any 12-month window within ongoing therapy?
Author response: These are not new users, but we control for years of therapy in the logistic model
Address data limitations: Consider elaborating more critically on how refill data may misclassify adherence.
Author response: We acknowledge this limitation that how refill data could lead to misclassification if patients where they are adherent could be considered non adherent or vice versa. (line:374-375)
Expand on GBTM model diagnostics: Include more detailed fit statistics and possibly a sensitivity analysis comparing different group numbers.
Author response: We have used model adequacy criteria as per Nagin criteria explain this and how it compares group members and addresses this concern in method and result section. (line:162-173)
Discuss missing data handling: Were there missing data in clinical covariates, and how were they handled?
Author response: Since the data has been collected from EMR we didn’t face issues with missing
Enhance discussion of interventions: What kinds of interventions (e.g., educational, pharmacologic, digital health tools) are likely to be feasible and culturally appropriate for this population?
Author response:
We have added in discussion an educational intervention which is culturally sensitive to the cultural norms and address barriers specific to this population can be more influential with patients leading to behavioral change, better adherence and ultimately enhance health outcome. (line:359-362)
Improve statistical reporting: Tables could benefit from confidence intervals for key baseline statistics to help contextualize variability.
We agree and thanks for your comment
This study offers valuable insight into adherence patterns among Hispanic breast cancer patients using robust analytic methods.
We thank the reviewer for thoroughly reviewing the manuscript and providing suggestions for improvement.
Reviewer 2 Report
Comments and Suggestions for Authors
Thanks for the opportunity for me to review this interesting paper. It leverages an novel group-based trajectory model and focus on breast cancer patients, a clinically meaningful patients.
Comments:
Abstract: (1) Please clarify the definition of adherence: The abstract states that adherence was defined as PDC > 80%, but then incorporates monthly PDCs into GBTM. It would help to clarify whether the trajectories reflect continuous PDC trends over time or binary monthly adherence (2) GBTM Model Justification: More detail is needed on the selection of the final trajectory model. Why was the 3-group model selected over the 4- or 5-group models? Were model fit statistics (e.g., BIC, AIC, entropy) used? (3) Please refine grammar and sentence flow for readability, especially in the Results and Conclusion sections.
INTRODUCTION: (1) The introduction could benefit from adding the component of the clinical consequences and/or economic burden associated with nonadherence of OET among the breast cancer group. (2) The justification for using GBTM is sound, but its value over traditional PDC analyses could be more clearly stated. Please emphasize that GBTM allows for detection of distinct behavioral patterns over time, which is especially useful for tailoring adherence interventions in a diverse population. Minor: Clarity and Sentence Structure: Several sentences are fragmented or awkwardly structured, making the narrative harder to follow. Example: “Therefore, potentially sensitive to agents targeting the estrogen signaling pathway…” is a sentence fragment.
MATERIALS AND METHODS:
(1) Consistent fonts: 2.5 Adherence measurements->Adherence Measurements; 2.6 Statistical analysis->Statistical Analysis and others. Please check and confirm the consistency.
(2) In developing the group-based trajectory model, it said that a binary indicator of monthly PDC was incorporated into GBTM.
Have the author thought about using PDC as a continuous variable to include into GBTM?
See a prior example: Pan, Y.Y., Devabhakthuni, S., Cooke, C.E. and Slejko, J.F., 2025. Group-Based Trajectory Models to Evaluate the Association of Lipid Testing and Statin Adherence. Drugs-Real World Outcomes, 12(1), pp.75-81.
(3) 2.2 Study cohort, it mentioned that those women who discontinued OET due to side effects were excluded.
However in the Binary adherence of Adherence measurements, it mentioned that Discontinuation of OET due to clinical progression or life threatening adverse effects was not considered nonadherent.
Could the author clarify this by providing clear clarification about eligibility criteria??
RESULTS
It looks good overall but the some typos in wording and consistency is required for the titles of table.
DISCUSSION
Comments: (1) The authors identified three adherence trajectories using GBTM, whereas prior studies have typically reported five to six trajectories. While the manuscript mentions that this may be due to the smaller sample size and shorter follow-up duration, it would strengthen the paper to more explicitly discuss the model selection criteria (e.g., BIC, AIC, entropy) that guided the final choice of three trajectories. (2) The comparison between GBTM and the PDC binary method is well presented. However, the discussion could be expanded to emphasize the clinical relevance of patients with borderline PDC scores (e.g., 0.724–0.799). Clarifying how these near-threshold patients might be managed differently in clinical settings would enhance the practical implications of the findings. (3) The authors rightly highlight the lack of similar studies in underserved populations. It may be helpful to elaborate on specific socioeconomic or systemic barriers affecting adherence in this group and how targeted interventions could be developed based on trajectory analysis.
Author Response
Reviewer 2:
We thank the reviewer for thoroughly reviewing the manuscript and providing suggestions for improvement. Point-wise response to each comment is given below. Corresponding changes in the manuscript are highlighted by using track changes.
Thanks for the opportunity for me to review this interesting paper. It leverages an novel group-based trajectory model and focus on breast cancer patients, a clinically meaningful patients.
Comments:
Abstract: (1) Please clarify the definition of adherence: The abstract states that adherence was defined as PDC > 80%, but then incorporates monthly PDCs into GBTM. It would help to clarify whether the trajectories reflect continuous PDC trends over time or binary monthly adherence (2) GBTM Model Justification: More detail is needed on the selection of the final trajectory model. Why was the 3-group model selected over the 4- or 5-group models? Were model fit statistics (e.g., BIC, AIC, entropy) used? (3) Please refine grammar and sentence flow for readability, especially in the Results and Conclusion sections.
Author response: we added a sentence for binary indicator of adherence in the abstract, GBTM justification details updated in the method but cannot be added in the abstract due to word limitation of 250 words. (line:22)
INTRODUCTION: (1) The introduction could benefit from adding the component of the clinical consequences and/or economic burden associated with nonadherence of OET among the breast cancer group. (2) The justification for using GBTM is sound, but its value over traditional PDC analyses could be more clearly stated. Please emphasize that GBTM allows for detection of distinct behavioral patterns over time, which is especially useful for tailoring adherence interventions in a diverse population. Minor: Clarity and Sentence Structure: Several sentences are fragmented or awkwardly structured, making the narrative harder to follow. Example: “Therefore, potentially sensitive to agents targeting the estrogen signaling pathway…” is a sentence fragment.
Author response: Thanks for the insights on introduction. We have added the sentence for GBTM and updated the introduction as suggested. Revised the grammatical errors. (line:49-50 and 81-83)
MATERIALS AND METHODS:
- Consistent fonts: 2.5 Adherence measurements->Adherence Measurements; 2.6 Statistical analysis->Statistical Analysis and others. Please check and confirm the consistency.
Author response: Updated the font and make it consistent through out
(2) In developing the group-based trajectory model, it said that a binary indicator of monthly PDC was incorporated into GBTM.
Have the author thought about using PDC as a continuous variable to include into GBTM?
See a prior example: Pan, Y.Y., Devabhakthuni, S., Cooke, C.E. and Slejko, J.F., 2025. Group-Based Trajectory Models to Evaluate the Association of Lipid Testing and Statin Adherence. Drugs-Real World Outcomes, 12(1), pp.75-81.
Author response: thank you for this insightful comment, we choose binary indicator of PDC as 0.8 cutoff of PDC has been shown in previous literature and widely accepted for endocrine therapy adherence and shown clinically meaningful adherence patterns.
Lailler G, Memoli V, Benjamin CLB, et al. Five-year adjuvant endocrine therapy adherence trajectories among women with breast cancer: a nationwide French study using administrative data. Clinical breast cancer. 2021;21(4):e415-e426.
(3) 2.2 Study cohort, it mentioned that those women who discontinued OET due to side effects were excluded.
However in the Binary adherence of Adherence measurements, it mentioned that Discontinuation of OET due to clinical progression or life threatening adverse effects was not considered nonadherent.
Author response: Yes per exclusion criteria these patients excluded from the study and we remove the second sentence from method section to avoid confusion.
Could the author clarify this by providing clear clarification about eligibility criteria??
RESULTS
It looks good overall but the some typos in wording and consistency is required for the titles of table.
Author response: we rephrased titles of tables for consistency
DISCUSSION
Comments: (1) The authors identified three adherence trajectories using GBTM, whereas prior studies have typically reported five to six trajectories. While the manuscript mentions that this may be due to the smaller sample size and shorter follow-up duration, it would strengthen the paper to more explicitly discuss the model selection criteria (e.g., BIC, AIC, entropy) that guided the final choice of three trajectories.
Author response: thanks for the great suggestion we have added “To determine the optimal number of adherence trajectories, we compared models 2-5 based on BIC of where smaller absolute values represent better model fit, clinical relevance and a minimum sample size of 5% required in each trajectory group. In addition, we have used Nagin model adequacy criteria and assessed the goodness of fit measures, including AvePP≥ 0.85, and OCC values > 5.0 for each trajectory, which indicates good classification accuracy” (line:320-326)
(2) The comparison between GBTM and the PDC binary method is well presented. However, the discussion could be expanded to emphasize the clinical relevance of patients with borderline PDC scores (e.g., 0.724–0.799). Clarifying how these near-threshold patients might be managed differently in clinical settings would enhance the practical implications of the findings.
Author response: 0.724–0.799 indicating a borderline adherence status which is closer to (PDC≥0.8) cutoff. In the clinical setting, these near-threshold patients could be managed well to prevent further decline of their adherence. (line:298-299)
(3) The authors rightly highlight the lack of similar studies in underserved populations. It may be helpful to elaborate on specific socioeconomic or systemic barriers affecting adherence in this group and how targeted interventions could be developed based on trajectory analysis.
Author response: We have elaborated on specific socioeconomic or systemic barriers affecting adherence in underserved populations and added in discussion that an educational intervention which is culturally sensitive to the cultural norms and address barriers specific to this population can be more influential with patients leading to behavioral change, better adherence and ultimately enhance health outcome. (line:355-362)
Reviewer 3 Report
Comments and Suggestions for Authors
Dear Authors,
Thank you for the opportunity to review your manuscript, “Group-Based Trajectory Model to Assess Adjuvant Endocrine Therapy Adherence Pattern in HR-positive Breast Cancer: Results from Rio Grande Valley Patients.” This is an important study that addresses a significant gap in Cancer care research.
As I reviewed the manuscript, I realized that the following suggestions would help improve the clarity and overall impact of the manuscript.
- In the abstract section, it is better to include the odds ratio for the important predictors – diabetes, years of therapy, etc., to strengthen the interpretation of the findings.
- The trajectory groups' final model percentage calculation needs revision. (236 – 244).
- Maintain consistency in labeling the proportion, in some places 43% (line – 275) and some 41% (line – 285). Maintain consistency throughout the manuscript.
- Avoid restating detailed findings again in the conclusion section.
Overall, I think this is a well-designed study with strong contextualization of findings with literature.
I appreciate the effort your team has put into it, and look forward to seeing the revised version.
Thank you.
Author Response
Author response: We thank the reviewer for thoroughly reviewing the manuscript and providing suggestions for improvement. Point-wise response to each comment is given below. Corresponding changes in the manuscript are highlighted by using track changes.
Reviewer 3:
Dear Authors,
Thank you for the opportunity to review your manuscript, “Group-Based Trajectory Model to Assess Adjuvant Endocrine Therapy Adherence Pattern in HR-positive Breast Cancer: Results from Rio Grande Valley Patients.” This is an important study that addresses a significant gap in Cancer care research.
As I reviewed the manuscript, I realized that the following suggestions would help improve the clarity and overall impact of the manuscript.
- In the abstract section, it is better to include the odds ratio for the important predictors – diabetes, years of therapy, etc., to strengthen the interpretation of the findings.
Author response: we added odds ratio as suggested (line:31-32)
- The trajectory groups' final model percentage calculation needs revision. (236 – 244).
Author response: we have updated the percentage in the table and through the manuscript, in the figure for its auto generated by SAS. (line 254-263)
- Maintain consistency in labeling the proportion, in some places 43% (line – 275) and some 41% (line – 285). Maintain consistency throughout the manuscript.
Author response: Updated the percentage to 43.9% though out the manuscript
- Avoid restating detailed findings again in the conclusion section.
Author response: We have removed the detailed findings from the conclusion section (line: 387-389)
Overall, I think this is a well-designed study with strong contextualization of findings with literature.
I appreciate the effort your team has put into it, and look forward to seeing the revised version.
Thank you.